# A Content Analysis of Persuasive Appeals Used in Media Campaigns to Encourage and Discourage Sugary Beverages and Water in the United States

**DOI:** 10.3390/ijerph20146359

**Published:** 2023-07-13

**Authors:** Vivica I. Kraak, Adrienne Holz, Chelsea L. Woods, Ann R. Whitlow, Nicole Leary

**Affiliations:** 1Department of Human Nutrition, Foods, and Exercise, Virginia Polytechnic Institute and State University (Virginia Tech), Blacksburg, VA 24061, USA; nleary@vt.edu; 2School of Communication, Virginia Polytechnic Institute and State University (Virginia Tech), Blacksburg, VA 24061, USA; holz@vt.edu (A.H.); clwoods@vt.edu (C.L.W.); awhitlow1975@vt.edu (A.R.W.)

**Keywords:** media campaigns, persuasive appeals, graphic images, sugary beverages, water, content analysis, United States

## Abstract

The frequent consumption of sugary beverages is associated with many health risks. This study examined how persuasive appeals and graphics were used in different media campaigns to encourage and discourage sugary beverages and water in the United States (U.S.) The investigators developed a codebook, protocol and systematic process to conduct a qualitative content analysis for 280 media campaigns organized into a typology with six categories. SPSS version 28.0 was used to analyze rational and emotional appeals (i.e., positive, negative, coactive) for campaign slogans, taglines and graphic images (i.e., symbols, colors, audiences) for 60 unique campaigns across the typology. Results showed that positive emotional appeals were used more to promote sugary beverages in corporate advertising and marketing (64.7%) and social responsibility campaigns (68.8%), and less to encourage water in social marketing campaigns (30%). In contrast, public awareness campaigns used negative emotional appeals (48.1%), and advocacy campaigns combined rational (30%) and emotional positive (50%) and negative appeals (30%). Public policy campaigns used rational (82.6%) and positive emotional appeals (73.9%) to motivate support or opposition for sugary beverage tax legislation. Chi-square analyses assessed the relationships between the U.S. media campaign typology categories and graphic elements that revealed three variables with significant associations between the campaign typology and race/ethnicity (χ^2^(103) = 32.445, *p* = 0.039), content (χ^2^(103) = 70.760, *p* < 0.001) and product image (χ^2^(103) = 11.930, *p* = 0.036). Future research should examine how positive persuasive appeals in text and graphics can promote water to reduce sugary beverage health risks.

## 1. Introduction

Extensive evidence shows that the habitual consumption of sugary beverage products is strongly associated with many health risks, including dental caries, obesity, type 2 diabetes and cardiovascular disease, for populations globally [1,2]. The World Health Organization (WHO) [3,4] and United States (U.S.) expert committees [5,6,7] have recommended that governments implement comprehensive policies, systems and environmental change strategies to restrict the availability, affordability, access and marketing of sugary beverages to reduce the diet and health risks of infants, toddlers, children, adolescents and adults. Creating environments that socially normalize safe potable water and healthy hydration behaviors is essential to promote optimal diets and health for populations [8].

In 2019, the United Nations (UN) Food and Agricultural Organization (FAO) and the WHO jointly released guidelines for Member State governments that encouraged water as the healthy default beverage as one of sixteen principles to support sustainable healthy diets [9]. The WHO does not recommend artificially sweetened, low- or no-calorie beverages for populations due to the potential adverse effects of long-term use, such as increased NCD and mortality risks in adults [10]. 

The Dietary Guidelines for Americans (DGA) 2020–2025 recommend that individuals consume water and other nutrient-dense, unsweetened beverages (i.e., low-fat and nonfat milk or a fortified nondairy soy milk and limited amounts of 100% juice and unsweetened coffee, tea and flavored water) to achieve a healthy weight and reduce diet-related health risks [11]. Both the DGA 2020–2025 report [11] and a scientific panel of the American Heart Association [12] also recommend that individuals reduce sugary beverage intake without substituting artificially sweetened, low- and no-calorie beverages, and that plain or carbonated unsweetened water should be the primary beverage of choice to achieve healthy hydration. 

### 1.1. Persuasive Communications Used in Media Campaigns 

Media psychology is a discipline that explores the influence of print, broadcast and digital media on people’s attitudes, perceptions, emotions, beliefs and behaviors [13]. Persuasive communications involve a conscious effort to influence a receiver using messages to shape, reinforce or change a behavior or response of an individual or population [14]. Transnational food, beverage and restaurant industry firms have used persuasive communications as part of integrated marketing communications (IMC) used to influence the attitudes, perceptions and behaviors of targeted populations [15]. Persuasive appeals are combined with many IMC components, which represent the strategies, techniques, channels and platforms used across diverse settings and channels to market branded food and beverage products to populations [15]. 

Taglines and slogans are “phrases singled out for emphasis in advertisements” that seek to communicate a brand’s distinctive identity, establish an emotional connection and reflect a need or benefit [16]. Taglines and slogans emphasize one or more textual messages and are often combined with graphics (i.e., images, colors, culturally relevant symbols and icons) or audio (i.e., jingles or music) to build brand awareness, memorability, trust and loyalty to influence a target audience’s knowledge, attitudes, beliefs, intentions, values and behaviors [17,18,19,20,21,22,23]. 

Persuasive communications are based on Aristotle’s rhetorical triangle, which combines three types of appeals including: ethos (credibility), pathos (emotional) and logos (rational) [24]. Research suggests that food, beverage and restaurant firms have used emotional appeals extensively in print, broadcast and digital media across different cultures [25]. Moreover, businesses have used positive emotional appeals and multisensory marketing to inspire fun, pleasure and taste in television advertising content and packaging to encourage children and adolescents to request, purchase and consume branded, energy-dense and nutrient-poor food and beverage products linked to obesity [15,26].

### 1.2. Formative Research on U.S. Media Campaigns Used to Promote Beverages 

Recent research has identified different types of media campaigns used to advertise and market sugary beverage brands to Americans, compared with other campaigns to encourage water and discourage sugary beverage consumption. Kraak and Consavage-Stanley (2021) published a formative paper that developed a theoretically grounded media campaign typology with six categories that differed by goal and paradigm [27] (Figure 1). 

Kraak et al. (2022) published a second paper that described a systematic scoping review that identified examples of U.S. media campaigns (*n* = 280) across the campaign typology (1886–2021) to promote or discourage sugary beverages or to encourage healthy beverages (i.e., water, low-fat milk and unsweetened tea, coffee and juice) [28]. Results showed that two-thirds of 280 U.S. media were corporate advertising or marketing used to promote branded sugary beverages (65.8%; *n* = 184) [28]. Less than 10 percent of the U.S. media campaigns were organized into the other five categories of the typology. The other campaigns were used to raise public awareness or education about the harms of sugary beverages (9.6%; *n* = 27); to mobilize individuals to support or oppose public policies such as sugary beverage taxes (8.2%; *n* = 23); to promote social marketing messages to select and drink water or other healthy beverages (7.1%; *n* = 20); to promote corporate social responsibility or public relations to influence how the public perceived businesses (5.7%; *n* = 16); and to change people’s views about sugary beverage firms through media advocacy and counter-marketing messages (3.6%; *n* = 10) [28]. 

Kraak et al. (2022) reported that out of the 280 media campaigns identified, only 24 evaluations for 20 unique beverage campaigns were implemented over 30 years (1992–2021) that had reported specific outcomes [28]. The results showed that the modestly funded social marketing and public health campaigns had changed the target audience’s short-term awareness, knowledge and consumption of water and low-fat milk and reduced sugary beverages. However, no media campaign had influenced the long-term social norm, policy or population health outcomes to reduce sugary beverage intake or encourage water and other healthy hydration behaviors [28]. These results are explained by many psychological, economic and multi-cultural factors that interact to influence the beverage preferences of individuals. Figure 2 shows illustrative examples of the graphic images and textual content used in a sub-set of the media campaigns (*n* = 60) across the six categories of the typology [28]. 

Public health practitioners have used print, broadcast and digital media platforms to raise awareness and influence the behaviors of diverse populations to reduce sugary beverage health risks [27,28]. Different types of media campaigns operate concurrently within complex IMC ecosystems, yet the content of the appeals used in textual messages and graphic images are rarely analyzed [27,28]. Kraak et al. (2022) identified six future research recommendations, including the need to explore how message framing, graphics, slogans and IMC strategies may influence media campaign outcomes [28]. This study addresses this research gap to understand how persuasive appeals have been used in different types of media campaigns to design effective communication strategies that discourage sugary beverages and encourage water to reduce sugary beverage health risks for populations. 

### 1.3. Study Purpose 

This study had two aims. The first was to examine how U.S. media campaigns have used persuasive emotional and rational appeals in slogans, taglines and graphic images to promote healthy and unhealthy non-alcoholic beverage products to Americans over decades. The second was to develop and apply a systematic process to examine the variance of persuasive appeals across different types of campaigns across the media campaign typology. 

This present study aimed to further build the evidence base by analyzing the nature and content of the persuasive appeals used in the slogans and taglines (*n* = 280) of a convenience sample and graphic images (*n* = 60) for the U.S. media campaigns identified through a systematic scoping review [28]. The results could inform future research to design effective persuasive communications to promote water and healthy hydration for Americans. 

## 2. Materials and Methods 

This study was guided by the three research questions (RQs) described below. 

RQ1: To what extent were persuasive appeals used in U.S. media campaign slogans and taglines across the six categories of the campaign typology to encourage or discourage sugary beverages or to encourage water and other healthy beverages to Americans? 

RQ2: To what extent were persuasive appeals used in U.S. media campaign slogans and taglines across the six categories of the campaign typology to encourage water and other healthy beverages and to discourage sugary beverage brands or products to Americans? 

RQ3: To what extent were graphic images used in a selected sample of U.S. campaigns across the six categories of the typology to influence the beverage behaviors of Americans? 

Persuasive appeals were defined as rational, emotional or combined to reach targeted audiences. Rational appeals in messages use logic, reasoning or factual information to influence the cognitive, affective or behavioral outcomes of individuals or populations. Emotional appeals used in campaign messages may be positive (i.e., happiness, hope, humor or pleasure); negative (i.e., fear, guilt, anger or risk); or coactive (i.e., hope and illness). Rational and emotional appeals influence a change in a person’s cognition (i.e., way of thinking and beliefs) that can result in a behavior change that becomes a habit through repetition [18,29,30]. 

The graphic content was defined as including images, symbols, icons and colors used in media campaigns to complement the content of persuasive appeals used in slogans, taglines and messages to increase the attention, interest, engagement and message recall of target audiences [31]. The effectiveness of persuasive appeals used in media campaigns to reduce sugary beverage health risks, or to gain support for sugary beverage tax policies, depends on the product or brand, cultural context, message content and target audience characteristics such as knowledge, perceived health-risk susceptibility, political orientation and media literacy [27,28,29,32,33,34,35]. 

This study used a directed qualitative content analysis to examine an existing database of 280 media campaigns [28]. This type of content analysis is used to examine a purposive sampling of content to understand a social reality [36,37]. Quantitative content analysis has been used to examine people’s exposure to food and beverage brands or products in different geographic locations and through social media platforms [38,39,40]. Some researchers have used automated content and discourse or sentiment analysis to examine large amounts of social media text messages, voice audio or image datasets [41,42]. We selected a directed qualitative content analysis to explore the existing convenience sample of 280 media campaigns identified across the six categories of media campaign typology. The coding methodology and data analysis are described below. 

### 2.1. Coding Methodology 

The coding methodology was guided by a directed qualitative content analysis [43] and used a purposive sample of previously published media campaigns [28]. Qualitative researchers recommend combining deductive and inductive approaches to explore theories and emerging concepts [44,45]. The directed approach is both deductive, where the codes are defined prior to and during analysis via existing research, and inductive, where codes are generated based on observations [36,46]. 

To answer RQ1 and RQ2, the unit of analysis was the individual campaign slogan or tagline. The categories of analysis were determined a priori, and the content was present and observable at the surface level [47]. We developed an initial codebook using preexisting categories of emotional and rational appeals based on the published literature [18,29,48]. CW led the process to construct the codebook, and all co-investigators participated in a systematic and iterative approach to examine the relevant literature on rational and emotional message appeals and to code for the presence or absence of each appeal [49,50]. This step was guided by the published literature for qualitative content analyses that examined advertising campaigns for their persuasive appeal content, graphic or visual images or people, products or brands, settings and colors [51]. The co-investigators met several times to develop a distinct and mutually exclusive definition for each appeal that was then used by two students (AW and NL) to code for the presence of both positive and negative emotions as a coactive appeal. 

To answer RQ3, the unit of analysis was the graphic image used in individual campaigns derived from an illustrative sample of images selected across the media campaign typology (Figure 2) [28]. The media categories 1–2 were selected to represent maximum diversity of corporate advertising or marketing campaigns that depicted beverages (i.e., soda and water); time frame (i.e., 1980s to present); industry firm (i.e., Coca-Cola Company, PepsiCo, Danone, Nestle and the ABA); and purpose (i.e., advertising, public relations and corporate social responsibility). The media categories 3–6 were selected to depict maximum diversity of social marketing, public information, media advocacy and public policy campaigns launched across U.S. geographic locations, and depicted racial, ethnic and gender diversity [28]. We developed an initial list of codes using preexisting categories of visual elements found in advertising and social marketing research for composition (i.e., image, slogan, setting and colors) [19,29,51,52,53,54,55]. Then, we conducted a latent analysis to determine the implied meaning using emergent codes generated from the media content [37,47]. 

After completing the initial codebook, the co-investigators held a training session to discuss the definition of each code and the coding process. Two co-investigators (AW and NL) coded a sample of taglines, slogans and images used for tobacco campaigns organized across the six categories of media campaign typology. Thereafter, AW and NL independently coded a second sample of tobacco-use campaigns. All authors met to discuss the results to resolve discrepancies in the coding process. After deliberations, the authors refined the codebook by combining codes that had similar meanings and overlapped and by removing less relevant codes. Recognizing that these categories were not exhaustive [44], we used inductive category development to add new codes that captured other key elements of the slogans/taglines and graphic images [36,55]. To enhance reliability, AW and NL independently coded a third sample of tobacco campaigns, and all authors met virtually to finalize the codebook. 

The next step involved the systematic selection of a random sample of 10 percent of the campaign slogans, taglines and images to create a sample for the intercoder reliability (ICR) analysis based on Lombard et al. (2002) [56]. Two co-investigators (AW and NL) independently coded 28 randomly selected slogans or taglines and 12 randomly selected graphic images used in various media campaigns to calculate the ICR for the coding categories using Krippendorff’s α [57]. 

After the first round of ICR, AW and NL independently coded the entire sample of 280 campaigns listed in Appendix A adapted from Kraak et al., 2022 [28]. Each coder was randomly assigned half of the content to code, with some overlap in cases for the ICR calculations. After coding was complete, a second round of ICR calculations was conducted. Following the two ICR rounds, we excluded five persuasive appeals, which included the rational appeal for solution and four positive emotional appeals (i.e., happiness, pleasure, hope and pride), that did not make the ICR cut-off threshold. Due to the exploratory nature of this study, we capped the reliability threshold at 0.65 to include in the final analysis. Although there are no established standards for acceptable intercoder reliability coefficients [56], we used this cutoff based on Krippendorff’s suggestion that α ≥ 0.667 may be acceptable for exploratory research and for making tentative conclusions [57]. Table 1 summarizes the two rounds of ICR scores for the rational and emotional message appeals and visual elements for 10 percent of the 280 media campaigns. 

### 2.2. Data Analysis

Appendix A presents the codebook used for the RQ1, RQ2 and RQ3 content analyses. The codebook and reliability measures were adapted from Casais and Pereira (2021) [18]. For the RQ3 codebook, graphic images included icons, symbols, colors and the characteristics of individuals depicted by race, ethnicity, sex, gender and age in selected print media campaigns shown in Figure 2. 

Appendix A lists the 280 U.S. media campaigns analyzed for RQ1 and RQ2, and 60 unique media campaigns’ print images identified from a list of 86 campaigns analyzed for RQ3. Figure 2 depicts the campaigns across a media campaign typology that was compiled through an iterative Google browser search between June 2021 and July 2022 [28]. 

To address RQ1 and RQ2, each campaign slogan or tagline was coded for the presence (*n* = 1) or absence (*n* = 0) of each appeal or graphic image, which were mutually exclusive, and verified independently by three co-investigators (VIK, AH and CLW). To address RQ3, AW and NL coded for the presence (*n* = 1) or absence (*n* = 0) of specific features of the campaign image, and independently coded 103 images across 60 unique media campaigns, totaling 86 campaigns since several campaigns (i.e., Rethink Your Drink and Pouring on the Pounds) were used in different locations over time. The Statistical Package for Social Sciences (SPSS) version 28.0 [58] was used to examine the frequency (number and prevalence) of the rational and emotional appeals for each typology category, and is summarized in the tables. The analysis was completed by January 26, 2023. 

## 3. Results 

RQ1 examined the extent to which the rational and emotional appeals (i.e., positive, negative or coactive) were used in the U.S. media campaign slogans and taglines to encourage or discourage sugary beverages or encourage water and other healthy beverages to Americans. Table 2 shows the frequency of rational and emotional message appeals for the U.S. media campaign slogans or taglines to answer RQ1. Three rational message appeals that met the ICR threshold for the analysis were comparative, factual and scarcity. Ten emotional appeals that met the ICR threshold were analyzed, including four positive emotional appeals (i.e., humor, relaxed, sexual and social); five negative emotional appeals (i.e., anger, disgust, fear, guilt and worry); and one coactive emotional appeal. 

Table 2 shows that the rational appeals were used in a quarter of the media campaigns with a greater frequency of factual rather than comparative appeals that compared one or more specific attributes of a beverage. In contrast, emotional appeals were used in 62.5%, the majority of the media campaign slogans and taglines, especially positive emotional appeals that highlighted social interactions and relaxation. Negative emotional appeals (6.1%) and coactive appeals (1.1%) were rarely used to encourage or discourage sugary beverages or encourage water, milk or 100% juice. 

RQ2 examined the extent to which persuasive appeals varied across the six categories of the campaign typology. Table 3 shows the frequency of rational and persuasive emotional appeals used in the 280 media campaigns across the six categories of media campaign typology. The results show that the corporate advertising and marketing campaigns (category 1) and corporate social responsibility and cause-marketing campaigns (category 2) used positive emotional appeals more frequently (64.7% and 68.8%, respectively) to promote sugary beverage brands and increase corporate legitimacy and public trust, compared with the corporate-sponsored social marketing media campaigns (30%) (category 3) to promote branded water. 

Table 3 reveals that government and civil society organizations have used negative emotional appeals in public information media campaigns (category 4) more frequently (48.1%) compared with rational and positive emotional appeals. Media advocacy campaigns (category 5) used a combination of appeal strategies, including rational (30%), positive emotional (50%), negative emotional (30%) and coactive (20%). Political and public policy campaigns (category 6) used rational (82.6%) and positive emotional appeals (73.9%) to motivate people to support or oppose sugary beverage tax legislation in several U.S. cities and states. Figure 3 visually displays the frequency of rational versus emotional (positive, negative and coactive) appeals across the six categories of media campaign typology. 

Table 4 shows the frequency of graphic and visual elements for the content analysis of a subset of 60 unique U.S. media campaigns used to encourage or discourage sugary beverages or encourage water and other healthy beverages to Americans across the six categories of media campaign typology. 

After calculating frequencies, chi-square analyses were used to assess the relationships between the U.S. media campaign typology categories and graphic elements. Analyses revealed three significant associations, which included campaign typology and race/ethnicity (χ^2^(103) = 32.445, *p* = 0.039), campaign typology and content (χ^2^(103) = 70.760, *p* < 0.001) and campaign typology and product image (χ^2^(103) = 11.930, *p* = 0.036). 

Table 4 shows that corporate social responsibility, public relations or cause-marketing campaigns (81.8%), public information campaigns (67.7%) and social marketing campaigns (64.3%) were more likely to not depict people in the graphic images. Corporate marketing campaigns were more likely to depict white individuals (26.5%), whereas public information campaigns were more likely to contain unclear depictions of racially and ethnically diverse individuals (16.2%) and black individuals (13.5%). Additionally, individuals were more likely to encounter positive messaging via corporate marketing campaigns (94.1%) and corporate social responsibility, public relations or cause-marketing campaigns (72.7%). Social marketing campaigns were more likely to present unclear content (72.7%), and public information campaigns were more likely to deliver negative content (48.6%). 

Corporate marketing campaigns (79.4%), public information campaigns (70.3%) and media advocacy campaigns (60%) were more likely to feature a product image, whereas corporate social responsibility campaigns (63.6%) and social marketing campaigns (57.1%) were less likely to include a product image. 

## 4. Discussion

This exploratory study was the first of its kind to conduct a qualitative content analysis of the textual slogans and taglines for 280 U.S. media campaigns used to promote or discourage beverages. The campaigns were identified through a systematic scoping review that used a theoretically grounded media campaign typology [28]. This study examined how different media campaign slogans and taglines (*n* = 280), and a selected sample of graphic images (*n* = 60), were used to promote or discourage sugary beverages and encourage water or other healthy beverages.

This study found that positive emotional appeals (especially those that fostered social interactions, social activities and relaxation) were used most frequently by beverage firms, in contrast to the rational, negative emotional or mixed appeals used in other types of media campaigns across the typology. Hornick et al. 2016 [59] described the concept of a “hierarchy of appeals” whereby positive emotional appeals may be more effective than other types of appeals. Although we did not examine the effectiveness of the different appeals, the frequency was higher for corporate advertising and marketing campaigns and corporate social responsibility and public relations campaigns that used positive emotional appeals to promote primarily sugary beverage brands (64.7%; *n* = 119) and increase corporate legitimacy and public trust in their business practices (68.8%; *n* = 11). 

Our findings support other published studies that have found food, beverage and restaurant firms have used positive emotional appeals in textual slogans and taglines and images to encourage children and teens to request, purchase and consume unhealthy food and beverage products through television advertising and product packaging [15,19,26,53,60]. Beverage firms have also partnered with U.S. celebrity athletes and entertainers to convey positive emotional appeals for branded sugary beverages, energy drinks and sports drinks to influence multicultural Latinx and black children and teens [61,62].

Corporate marketing campaigns (79.4%), public information campaigns (70.3%) and media advocacy campaigns (60%) were more likely to feature a product image, whereas corporate social responsibility campaigns (63.6%) and social marketing campaigns (57.1%) were less likely to include a product image. 

Our study also showed that corporate marketing campaigns were more likely to depict white individuals (26.5%) whereas public information campaigns were more likely to show an unclear race or ethnic depiction (16.2%) and/or black individuals. 

In contrast, the social marketing campaigns used rational appeals (55%; *n* = 11) and positive emotional appeals (30%; *n* = 6); however, they infrequently used negative emotional appeals (5%; *n* = 1) in the textual messages. Social marketing campaigns were more likely to present unclear content (72.7%), and public information campaigns were more likely to deliver negative content (48.6%). The public information campaigns used negative emotional appeals (i.e., fear, guilt, worry, disgust) more frequently (48.1%; *n* = 13) to discourage sugary beverage intake. The corporate social responsibility, public relations or cause-marketing campaigns (81.8%), public information campaigns (67.7%) and social marketing campaigns (64.3%) were more likely to *not* depict people in the graphic images. 

Media advocacy and counter-marketing campaigns combined positive emotional appeals (50%; *n* = 5); rational and negative emotional appeals (30% each; *n* = 3); and two campaigns (i.e., The Bigger Picture (2013) and Share a Coke with Obesity (2015)) combined positive and negative emotional appeals in coactive appeals (20%; *n* = 2). Political or public policy campaigns used rational appeals (82.6%; *n* = 19) and positive emotional appeals (73.9%; *n* = 17), but not coactive emotional appeals to urge citizen support or opposition for sugary beverage tax legislation in U.S. cities (i.e., Albany, Berkeley, Oakland and San Francisco, California) and in California, Oregon and Washington states.

Our study findings did not show significant differences in the use of color in media campaigns. However, research has shown that visual images enhance the salience of issues to generate stronger framing effects on people’s emotions, opinions and behaviors compared with only text [31]. Graphic design and colors have been used strategically by marketers as branding elements that combine logos, fonts, slogans, taglines and trademarks [26,53]. Sugary beverage and energy drink firms have used colors and visual design strategically in social media posts to attract consumers’ attention to differentiate and build loyalty for their brands in a highly competitive marketing environment [53,61,62]. People across cultures have different preferences and meanings for colors and color combinations [63]. Marketers use color to convey meaning, reinforce cultural traditions, encourage patriotism, and to enable consumers to distinguish brands and products that are protected by trademark law [63]. 

### 4.1. Future Research on Positive Emotional Appeals: Pleasure, Happiness, Hope and Pride

This exploratory study is the first of our knowledge that examined the content of many persuasive appeals and visual elements of media campaign slogans, taglines and images to inform future communication strategies that may discourage sugary beverages and encourage water and other healthy beverage choices. 

Some of the variables examined in this study did not achieve an adequate ICR (Krippendorff’s α above 0.65) between the two independent coders. Four variables for positive emotional appeals (i.e., happiness, hope, pleasure and pride) were excluded from the analysis. However, we believe that these are important constructs that deserve further examination in future research to understand how advertisers and marketers have used these persuasive appeals in advertising and marketing campaigns to influence the cognitive, emotional or affective and behavioral outcomes of targeted populations. Therefore, we provide a synthesis of the published evidence in the section below for researchers and staff in government and civil society advocacy groups to understand how beverage firms have used positive emotional appeals, such as pleasure and happiness, to influence social norms for brand and product loyalty, and aspirational behaviors and lifestyles. While these concepts are often used interchangeably, these constructs have distinct meanings. 

Bédard et al. (2020) described pleasure as a multi-faceted concept that includes sensory and social experiences, food and beverage characteristics (i.e., sensory attributes, preparation process, novelty), variety, memories and the places associated with consuming these products [64]. Pettigrew (2016) described pleasure as a promising but underutilized component in social marketing campaigns to encourage healthy eating [65]. Emotional appeals have been used to promote pleasure for adults to prefer milk, and rational appeals have been used to promote the health profile of milk to encourage consumption [66]. Persuasive appeal messages could highlight the immediate pleasure from consuming water for hydration (such as Seattle’s Be Ready, Be Hydrated campaign targeting multi-cultural youth) rather than focusing on the long-term health effects [67]. The initial codebook for this study defined pleasure as the “emotion or sensation induced by the enjoyment or anticipation of what is felt, viewed or experienced as good or desirable,” which included similar concepts of enjoyment, refreshment and taste [68]. 

Lustig (2017) [69] distinguished between pleasure and happiness based on the nature of an experience, and the psychological and physiological responses associated with positive or negative outcomes. Pleasure is described as “short-lived and visceral (i.e., felt by the body); usually experienced alone (i.e., eating, shopping, drinking or binging); and often experienced with substances.” Lustig [69] associated excessive pleasure with increased levels of the neurotransmitter dopamine in the human brain, which fuels desire, motivation and may lead to addictive behaviors. In contrast, happiness is defined as a long-lived, purposeful and meaningful experience, usually in social groups, such as spending time with family or friends. Happiness is associated with increased levels of the neurotransmitter serotonin in the human brain, which fosters satisfaction and contentment. Happiness cannot be achieved by consuming foods, beverages or alcohol products, and too little happiness may lead to depression [69]. 

PepsiCo and The Coca-Cola Company dominate the national and global marketplace for sugary beverage sales and have used advertising, marketing and entertainment media campaigns as part of a broader IMC approach to increase sales, revenue and brand equity [27,28]. Table 2, Figure 3 and Appendix A show that these beverage firms have used positive emotional appeals in slogans or taglines to depict relaxation (i.e., warmth and people enjoying relationships with family and friends) and social activities (i.e., people enjoying active lifestyles, parties or shared social experiences). Table 3 shows that beverage firms have used positive emotional appeals in advertising and marketing campaigns (119, 64.7%); corporate social responsibility, public relations and cause-marketing campaigns (11, 68.8%); and in political and policy campaigns to challenge sugary beverage tax legislation (17, 73.9%). Positive emotional appeals were used less often for social marketing campaigns to promote water and other healthy beverages (6, 30%), perhaps because these are less profitable than branded sugary beverages. 

The initial codebook defined hope as “wanting something to be true and usually having a good reason to think that it might be true” [70]. Hope included the concepts of aspiration, enthusiasm, motivation and optimism. The initial codebook defined pride as a “feeling of satisfaction or delight in something one has achieved and/or is able to do” [48] which may extend beyond a personal level to an organization, community or country [71]. Pride included the concepts of freedom, nationalism and patriotism. Formative research on sugary beverage marketing to multicultural populations suggests that beverage firms have used pleasure, happiness, hope, pride and humor as persuasive appeals to elicit positive emotional responses from targeted audiences to increase brand loyalty, sales and the consumption of many types of sugary beverage products [72,73]. 

### 4.2. Future Research on Negative Emotional Appeals: Anger, Fear and Guilt

This study found that the first two categories in the media campaign typology (i.e., advertising and marketing, public relations and corporate social responsibility) used primarily positive emotional appeals to reach audiences. Category three (social marketing campaigns) used primarily rational appeals to promote healthy beverages. The remaining types of media campaigns (i.e., public awareness and education, counter-marketing and media advocacy, and political or public policy) used negative emotional appeals, mixed negative and positive emotional or used rational appeals, and positive emotional appeals were used much less often. 

In this study, no campaign had used an anger appeal, and only a few counter-marketing or media advocacy campaigns had used fear (3.2%; *n* = 9) and guilt (3.9%; *n* = 11) as a negative persuasive appeal to influence the targeted populations (Table 2). One study that evaluated anger appeals to promote activism among parents revealed a low level of support for policies to restrict sugary beverage marketing to U.S. children [74]. Other studies have found that using fear or guilt appeals to discourage sugary beverage intake depends on several mediating factors, such as the context, strength of the argument, pre-existing beliefs of individuals about sugary beverage harm, target audience characteristics and message framing [75,76,77]. 

Some research suggests that negative persuasive messages that convey the harms of sugary beverages may elicit reactance or resistance from targeted groups [78,79]. Fear appeals used in social marketing campaigns may have weak or unanticipated effects or even produce heightened anxiety or complacency among targeted and non-targeted populations [78]. An evaluation of racially and ethnically diverse adults’ (*n* = 618) responses to the 2012 New York City Pouring on the Pounds public information media campaign found a reactance in a sub-population, which was attributed to prior message exposure, current sugary beverage consumption habits and the target audience’s political views [79]. A sugary beverage threat appeal may influence one’s attitudes to adopt healthier behaviors (categories 3 and 4 of the campaign typology) but reduce political or public policy support to restrict sugary beverage availability and access (categories 5 and 6 of the campaign typology) [79]. Peers and family members discussed the media campaign’s messages, and there were both direct and indirect counter-persuasive effects for choosing sugary beverages based on the quality and number of personal interactions [80]. 

### 4.3. How Findings May Inform Policies, Programs and Research to Promote Healthy Hydration

Several U.S. cities and institutions have implemented policies, systems and environmental change strategies over decades to discourage or restrict sugary beverage availability, access and affordability, and to promote water and other healthy hydration behaviors. These strategies include establishing healthy beverage standards in schools and child-care settings; enacting excise taxes to support community programs, infrastructure and make safe tap drinking water available; requiring chain restaurants to provide healthy default beverages for children’s meals; implementing public awareness campaigns about the health risks of sugary beverages; and enacting health warnings on beverage products at point-of-sale in retail settings [81,82].

Major beverage firms and the leading U.S. industry trade association, the American Beverage Association (ABA), have used corporate advertising, social responsibility and political media campaigns extensively to influence the U.S. public and decision makers to oppose legislation [81]. The ABA used media campaigns as part of a coordinated strategy for its member firms, PepsiCo and The Coca-Cola Company, to defeat New York City’s portion size cap rule in 2013, and to repeal San Francisco’s warning label for sugary beverage products in 2016 [81,82]. The ABA also launched a corporate social responsibility campaign called the 2025 Beverage Calories Initiative [83] to influence the public’s positive views about business actions framed as promoting “balanced lifestyles while protecting consumer choice” in selected U.S. communities [84].

Some researchers suggest that U.S. media campaigns that contain messages that discourage unhealthy sugary beverages may be more promising to improve healthy hydration compared with messages that only promote water [85]. Our study showed that corporate media campaigns have primarily used positive emotional appeals to market branded sugary beverages instead of rational or negative emotional appeals to discourage sugary beverages. Graphic warnings on sugary beverages are used in social media advocacy campaigns as a rational and/or negative fear appeal to discourage parents from purchasing these products for children. Future research should empirically test different types of graphic and textual warnings that do not exacerbate weight-related bias or create unintended consequences, such as eating disorders, among susceptible individuals [86]. Research could also expand the sample size of different media campaigns analyzed from different countries to examine how positive emotional appeals may promote water for healthy hydration in slogans or taglines, narratives and graphics that convey actionable, memorable and motivational content that encourages water and discourages sugary beverages [87]. 

Many media campaigns moved online during the coronavirus pandemic, and beverage firms began direct advertising and marketing of sugary beverage brands and products to ethnically and racially diverse teens and young adults through Meta’s Instagram, Amazon’s Twitch and Twitter [40,88,89]. Future research is needed to develop and test media content to encourage healthy and environmentally sustainable beverage behaviors (i.e., drinking clean tap water instead of buying commercial branded water in single-use plastic containers) associated with larger climate and water footprints [90]. 

### 4.4. Study Strengths and Limitations

The strengths of this study were the interdisciplinary approach that combined expertise from media psychology and communications, behavioral change, public health nutrition and policy to examine 280 beverage media campaign taglines, slogans and the graphic content for 60 campaigns using a novel media campaign typology. This study created a codebook, protocol and systematic process that was tested and could be refined for future research. The study limitations were that each construct examined is multi-dimensional. We excluded four positive persuasive appeals (i.e., happiness, pleasure, hope and pride) that did not receive adequate ICR values for the final analysis. The codebook that we developed could be expanded, tested and refined to guide future research on the content of media slogans and taglines within specific cultural and political contexts, and on other visual elements that influence consumers’ responses to text and visual images. 

## 5. Conclusions

Corporate advertising, marketing and public relations campaigns use positive emotional appeals and positive image content to influence beverage product awareness, preferences and behaviors, and to reinforce positive public perceptions of beverage firms. This study examined the persuasive rational and emotional appeals and images used in 280 U.S. media campaigns categorized across six distinct categories to promote healthy and unhealthy non-alcoholic beverage products to Americans over many decades. Government and civil society organizations should support future research to explore how positive emotional appeals may be used effectively in textual messages and graphic content to promote water as the healthy default beverage and to socially normalize water and healthy hydration behaviors to reduce sugary beverage health risks for Americans. 

## Figures and Tables

**Figure 1 ijerph-20-06359-f001:**
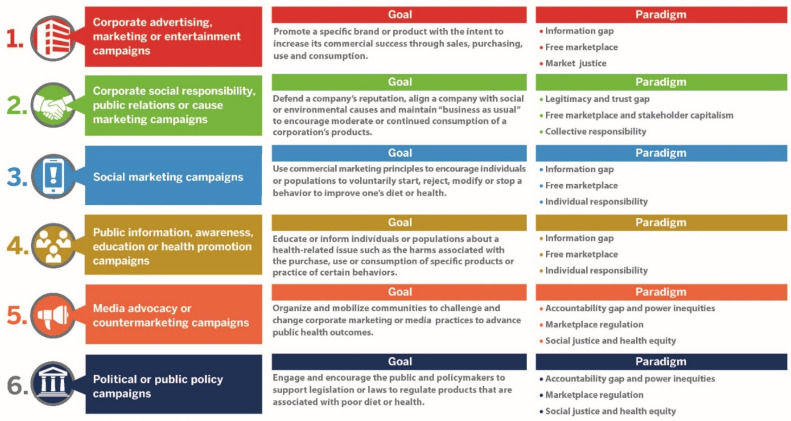
A media campaign typology defined by goal and paradigm. Reference [27].

**Figure 2 ijerph-20-06359-f002:**
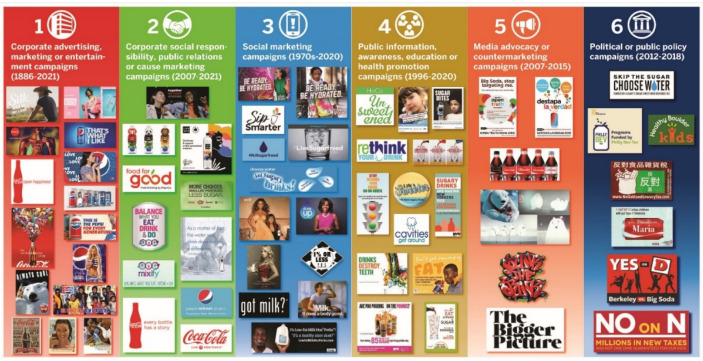
A media campaign typology with examples of taglines, slogans and images used to promote or discourage sugary beverages and encourage water, milk and juice to Americans, 1886–2021. Adapted from reference [28].

**Figure 3 ijerph-20-06359-f003:**
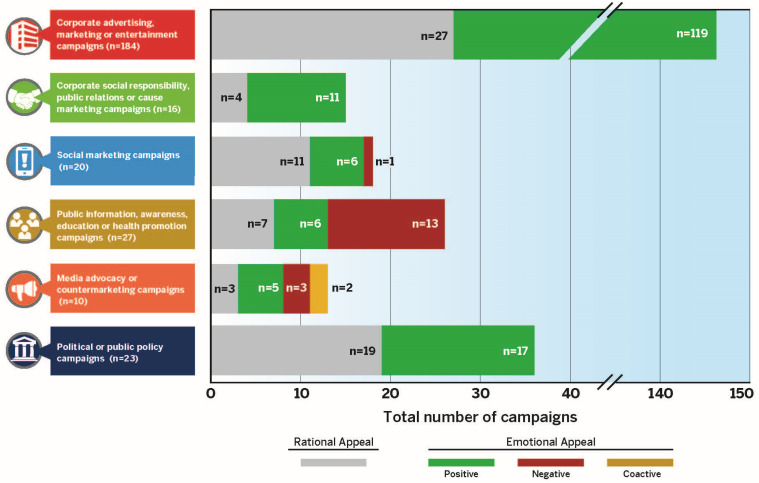
Frequency of rational and persuasive emotional appeals used in the 280 media campaigns across the six categories of media campaign typology. *Note:* The appeals were not mutually exclusive; therefore, the totals do not align with the number of media campaigns under each category.

**Table 1 ijerph-20-06359-t001:** Intercoder reliability (ICR) scores for the rational and emotional message appeals and the graphic and visual elements assessed for a sample of media campaigns used to encourage or discourage sugary beverages or encourage water and other healthy beverages to Americans.

Coding Category	ICR (Time 1)	ICR (Time 2)
Rational Message Appeals		
Rational	1.00	0.78 ^a^
Comparative	1.00	1.00
Factual	1.00	1.00
Scarcity	1.00	1.00
Solution	1.00	0.22 ^b^
Emotional Message Appeals		
Emotional	0.85	0.72 ^a^
Emotional (Positive)	0.86	0.79 ^a^
Happiness	0.63	0.27 ^b^
Hope	1.00	0.28 ^b^
Humor	1.00	1.00
Pleasure	0.92	0.31 ^b^
Pride	1.00	0.47 ^b^
Relaxed	1.00	1.00
Sexual	1.00	1.00
Social	1.00	0.65
Emotional (Negative)	1.00	0.65
Anger	1.00	1.00
Disgust	1.00	1.00
Fear	1.00	1.00
Guilt	1.00	0.65
Worry	1.00	1.00
Coactive	1.00	1.00
Graphic and Visual Elements		
People	1.00	1.00
Biological Sex	1.00	0.84
Age	1.00	1.00
Race/Ethnicity	1.00	1.00
Emotion	1.00	1.00
Content	0.76	0.86
Color	0.89	0.76
Product	1.00	1.00

^a^ Adjusted intercoder reliability score after removing variables with low reliabilities (α < 0.65). ^b^ Message appeals were excluded from final analysis because of low reliabilities (α < 0.65).

**Table 2 ijerph-20-06359-t002:** Frequency of rational and emotional message appeals for the content analysis of 280 U.S. media campaign slogans or taglines used to encourage or discourage sugary beverages or encourage water and other healthy beverages to Americans.

Coding Category	Frequency (n=, %)	
** *Rational Message Appeals* **		** *Examples* **
Rational	71, 25.4%	1% or Less, Dunk the Junk and Change the Tune
Comparative	7, 2.5%	Choose Water Not Sugary Drinks
Factual	17, 6.1%	Drinks Destroy Teeth
Scarcity	0, 0%	
** *Emotional Message Appeals* **		** *Examples* **
Emotional	175, 62.5%	
*Emotional (positive)*	164, 58.6%	Get Healthy Philly, Healthy for Good and Sip Smarter
Humor	0, 0%	
Relaxed	26, 9.3%	Have a Coke and Smile
Sexual	1, <1%	
Social	41, 14.6%	Share a Coke With, Together We Must, Coming Together: Translated and Keep Seattle Livable for All
*Emotional (negative)*	17, 6.1%	
Anger	0, 0%	
Disgust	6, 2.1%	Are You Pouring on the Pounds? and Soda Sucks
Fear	9, 3.2%	Sugar Bites
Guilt	11, 3.9%	Rethink Your Drink
Worry	7, 2.5%	Drinks Destroy Teeth
Coactive	2, <1%	Share a Coke With Obesity and The Bigger Picture

**Table 3 ijerph-20-06359-t003:** Frequency of rational and persuasive emotional appeals used in the 280 media campaigns across the six categories of media campaign typology.

Media Campaign Typology Category (% Campaigns; *n* = Number) of 280 Campaigns	Frequency of Appeals (n=, %)			
	Rational	Emotional		
		Positive	Negative	Coactive
1. Corporate advertising, marketing and entertainment campaigns (65.8%; *n* = 184)	27, 14.7%	119, 64.7%	0, 0%	0, 0%
2. Corporate social responsibility, public relations or cause-marketing campaigns (5.7%; *n* = 16)	4, 25%	11, 68.8%	0, 0%	0, 0%
3. Social marketing campaigns (7.1%, *n* = 20)	11, 55%	6, 30%	1, 5%	0, 0%
4. Public information, awareness, education or health promotion campaigns (9.6%, *n* = 27)	7, 25.9%	6, 22.2%	13, 48.1%	0,0%
5. Media advocacy or counter-marketing campaigns (3.6%; *n* = 10) *	3, 30%	5, 50%	3, 30%	2, 20% ***
6. Political or public policy campaigns (8.2%; *n* = 23) **	19, 82.6%	17, 73.9%	0, 0%	0, 0%

* Total of 13 persuasive appeals, because some media advocacy or counter-marketing campaign slogans or taglines were coded for rational and emotional appeals (positive, negative and coactive). ** Total of 36 persuasive appeals, because some political or public policy campaign slogans or taglines were coded as rational and emotional appeals such as: Keep Seattle Livable for All (Seattle, WA, USA) (factual and social). *** Two coactive appeals in category 5 were: The Bigger Picture (San Francisco, CA, 2013) and Share a Coke with Obesity (2015). The appeals were not mutually exclusive; therefore, the totals do not align with the number of media campaigns under each category.

**Table 4 ijerph-20-06359-t004:** Frequency of graphic and visual elements for the content analysis of 60 unique U.S. media campaigns used to encourage or discourage sugary beverages or encourage water and other healthy beverages to Americans.

Coding Category	Frequency Across Six Media Campaign Typology Categories (n=, %)					
	1. Corporate Advertising, Marketing and Entertainment Campaigns (*n* = 184)	2. Corporate Social Responsibility, Public Relations or Cause-Marketing Campaigns (*n* = 16)	3. Social Marketing Campaigns (*n* = 20)	4. Public Information, Awareness, Education or Health Promotion Campaigns (*n* = 27)	5. Media Advocacy or Counter-Marketing Campaigns (*n* = 10)	6. Political or Public Policy Campaigns (*n* = 23)
People	18, 52.9%	2, 18.2%	5, 35.7%	12, 32.4%	0, 0%	0, 0%
Biological Sex						
No individuals pictured	16, 47.1%	9, 81.8%	9, 64.3%	25, 67.6%	0, 0%	0, 0%
Unclear	2, 5.9%	1, 9.1%	0, 0%	3, 8.1%	0, 0%	0, 0%
Both male and female	7, 20.6%	0, 0%	1, 7.1%	0, 0%	0, 0%	0, 0%
Male	3, 8.8%	1, 9.1%	1, 71%	5, 13.5%	0, 0%	0, 0%
Female	6, 17.6%	0, 0%	3, 21.4%	4, 10.8%	0, 0%	0, 0%
Age						
No individuals pictured	16, 47.1%	9, 81.8%	9, 64.3%	25, 67.6%	0, 0%	0, 0%
Unclear	5, 14.7%	1, 9.1%	1, 7.1%	2, 5.4%	0, 0%	0, 0%
Multiple ages pictured	2, 5.9%	1, 9.1%	1, 7.1%	1, 2.7%	0, 0%	0, 0%
Elementary school-aged children	1, 2.9%	0, 0%	1, 7.1%	8, 21.6%	0, 0%	0, 0%
Teens/adolescents (up to age 18)	0, 0%	0, 0%	0, 0%	0, 0%	0, 0%	0, 0%
Adults (18 and older)	10, 29.4%	0, 0%	2, 14.3%	1, 2.7%	0, 0%	0, 0%
Race/Ethnicity *						
No individuals pictured	16, 47.1%	9, 81.8%	9, 64.3%	25, 67.6%	0, 0%	0, 0%
Unclear	5, 14.7%	0, 0%	1, 7.1%	6, 16.2%	0, 0%	0, 0%
Multiple races or ethnicities pictured	0, 0%	2, 18.2%	1, 7.1%	0, 0%	0, 0%	0, 0%
White	9, 26.5%	0, 0%	2, 14.3%	1, 2.7%	0, 0%	0, 0%
Black	4, 11.8%	0, 0%	1, 7.1%	5, 13.5%	0, 0%	0, 0%
Latinx	0, 0%	0, 0%	0, 0%	0, 0%	0, 0%	0, 0%
Native American or Pacific Islander	0, 0%	0, 0%	0, 0%	0, 0%	0, 0%	0, 0%
Asian	0, 0%	0, 0%	0, 0%	0, 0%	0, 0%	0, 0%
Emotion						
Not applicable	19, 55.9%	9, 81.8%	9, 64.3%	27, 73%	0, 0%	2, 100%
Positive	15, 44.1%	2, 18.2%	5, 35.7%	6, 16.2%	0, 0%	0, 0%
Negative	0, 0%	0, 0%	0, 0%	4, 10.8%	0, 0%	0, 0%
Content *						
Unclear	2, 5.9%	3, 27.3%	6, 42.9%	11, 29.7%	1, 20%	1, 50%
Positive	32, 94.1%	8, 72.7%	8, 57.1%	4, 10.8%	1, 20%	1, 50%
Negative	0, 0%	0, 0%	0, 0%	18, 48.6%	2, 40%	0, 0%
Coactive	0, 0%	0, 0%	0, 0%	4, 10.8%	1, 20%	0, 0%
Color						
Many or no dominant color	10, 29.4%	5, 45.5%	6, 42.9%	18, 48.6%	3, 60%	1, 50%
Blue	3, 8.8%	2, 18.2%	3, 21.4%	7, 18.9%	0, 0%	0, 0%
Brown	2, 5.9%	0, 0%	0, 0%	2, 5.4%	0, 0%	0, 0%
Gray	0, 0%	1, 9.1%	1, 7.1%	0, 0%	0, 0%	0, 0%
Green	1, 2.9%	1, 9.1%	0, 0%	1, 2.7%	0, 0%	0, 0%
Orange	0, 0%	0, 0%	0, 0%	1, 2.7%	0, 0%	0, 0%
Red	10, 29.4%	2, 18.2%	0, 0%	3, 8.1%	1, 20%	1, 50%
Violet	0, 0%	0, 0%	0, 0%	1, 2.7%	0, 0%	0, 0%
Yellow	0, 0%	0, 0%	0, 0%	1, 2.7%	0, 0%	0, 0%
Black	6, 17.6%	0, 0%	4, 28.6%	1, 2.7%	1, 20%	0, 0%
Traffic light color scheme	0, 0%	0, 0%	0, 0%	2, 5.4%	0, 0%	0, 0%
Patriotic color scheme	2, 5.9%	0, 0%	0, 0%	0, 0%	0, 0%	0, 0%
Product						
Absent	7, 20.6%	7, 63.6%	8, 57.1%	11, 29.7%	3, 60%	1, 50%
Present	27, 79.4%	4, 36.4%	5, 42.9%	26, 70.3%	2, 40%	1, 50%

* Chi-square test indicated significant difference (*p* < 0.05).

## Data Availability

Supporting data analyzed may be accessed at the weblink above in the Supplemental Materials.

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
