# Peer review of "A Content Analysis of Persuasive Appeals Used in Media Campaigns to Encourage and Discourage Sugary Beverages and Water in the United States"

_ijerph, 2023, doi:10.3390/ijerph20146359_

Round 1
Reviewer 1 Report
The authors clearly put a lot of effort into this thoughtful study. However, I have identified a number of issues that I believe should be addressed before this is published.
My biggest note is to make the study's "so what" much, much clearer earlier one - that is, make this more explicit in the Abstract and Intro. It took me a while to understand what value the authors saw in analyzing appeals and graphics in detail across the marketing categories. Why code that as opposed to other features within these campaigns? I'm frankly still not sure I understand the practical or theoretical implications of some of the findings - especially the findings related to graphics, which aren't really touched upon much in the Discussion. I think it would be helpful if the authors bring the summaries of relevant effects literature, which now are largely in the Discussion, up into the Introduction. A content analysis is only meaningful when it's coding something that clearly will have an important impact on a target population. The current Intro, which basically just says that campaigns use appeals, doesn't do enough to motivate or set up a content analysis that gets this in-depth. I also think the authors could push the envelope further in their interpretation of their findings to further drive home a "so what?". In the absence of more research, it seems reasonable for the various healthy campaign categories to make an effort to mirror approaches employed by the commercial campaigns--given the literature the authors review suggesting that the commercial campaigns are more effective.
Secondary notes:
- I'm confused about the graphics analyses. Why weren't graphics from all campaigns in the sample coded? Did this sub-sample contain graphics from 20 or 60 campaigns?--the former number is listed on page 3 and the latter on page 8.
- I feel like the "To what extent were..." framing of the RQs undersells the study. The study looks at what appeals and graphics were used and how usage rates vary across campaign types. That's richer than just tallying up the total number of appeals and graphics without further coding - which is what "To what extent" implies to me.
- Would it be possible to find a way to make some of the codes--or at least some of the more prevalent codes--more concrete in the body of the manuscript? What's an example of a "Social" appeal (etc)?
- The results would be easier to follow and the table easier to read if the authors ran a chi-square test (or similar) and included asterisks to help readers spot significant differences. Table 4 in particular is so massive that I'm not sure where to direct my attention--and the fact that these findings are barely mentioned in the Discussion doesn't help me identify key takeaways.
- I don't think you need to repeat the results in as much detail as you do in the Discussion. That kind of reporting really belongs just in the Results.
- The section on "positive emotional appeals" in the Discussion is confusing to me. Why would the paper dedicate so much space (about 1/3 of the Discussion) to findings that coders couldn't get reliable on and therefore didn't report on in the paper? I recommend cutting most of that section - perhaps bringing just the current final paragraph of that section into the intro somewhere.
Author Response
IJERPH Manuscript 2321332
A Content Analysis of Persuasive Appeals and Graphics for Media Campaigns used to Encourage and Discourage Sugary Beverages and Water in the United States
Please see the attachment.

Reviewer 2 Report
It is obvious that the brand wants to associate positively. The water, like other product, should try to associate itself positively.
Perhaps the issue here is not only advertising and persuasive communication, because water (its taste) is not as attractive as other drinks. It is also important to highlight the cultural factor of gastronomy (food and drink), since it is society that accepts the recommendation of advertising messages.
The research could have explicit data of the quantity of the analyzed sample where there are only rational messages, only emotional (positive and negative) and where they are mixed (rational and emotional -positve and negative-) and be able to observe the persuasive weight in this regard.
Some part of the text mix information not relevant to the subject of the investigation.
Author Response
Please see the attached comments in response to the revieer's points.

Reviewer 3 Report
Background: This is a very interesting research question and extremely relevant to the field of public health that often relies on media campaigns to promote health. Very creative and research driven use of the typology – this is very helpful in understanding the different categories of campaigns.
Materials and Methods: I thought methodology was backed up by strong evidence and I am impressed by the attention to detail. I have a few minor comments. Lines 202 -206 address RQ3, however, this section was very unclear to me the steps that were taken in this process. I think this paragraph could use some clarification for the readers unfamiliar with latent analysis/emergent codes. I also am curious as to who was involved in the coding methodology. The author states “we constructed the codebook..” Is this the authors? Graduate students? How many people were involved in this process? In paragraph that starts at line 207, this is done extremely well!
Results: Tables are clear and results written succinctly. I wonder if you could include a table or some examples of the codebook? It may help the reader understand more about the content of the persuasive appeals if you demonstrate exactly what these codes look like. For example… what would be an example of the rational coding categories (i.e. comparative, factual, scarcity, etc)? I do see you include the codebook in a supplemental file, is there a way to clearly reference this in the methodology section?
Discussion/conclusion: Well-researched and organized. No major comments here.
Author Response

(The authors gave the same response as above.)

Reviewer 4 Report
Paper "How Have Media Campaigns Used Persuasive Appeals to Encourage and Discourage Sugary Beverages and Water in the United States? A Content Analysis of 280 Campaigns to Inform Policy, Practice and Research to Reduce Sugary Beverage Health Risks" deals with examining appeals that affect consumers with the aim of reducingthe consumption of sugary beverages and increasing water and hydrating drinks.
The topic is very important from public health point of view. Increased consumption of beverages and less water and hydrating drinks consumption is a risky health-related behavior.
In order for public health actions to be effective, examination of appeals as important factors that influence consumers is necessary, with the aim of preventing abuse, but also using new knowledge to encourage positive health-related behavior.
This reviewer recommendation is to reduce the number of words in the title.
This reviewer suggests that the paper can be accepted after minor revision.
I have a few comments that can help improve the quality of the work:
1. line 125-131 Explain why it did not affect long-term changes in habits towards sugary beverage consumption. A psychological argument could be used.
2. line 132-137 Why are they focused exclusively on the reduction of sugary beverages and not on the promotion of water and hydrant drinks?
Research has shown that the influence of sugar on the brain can have an addictive effect and stimulate the secretion of dopamine. Maybe take that information into consideration?
3. line 153-155 Rephrase the sentence. Rational and emotional appeals influence a change in cognition (way of thinking, beliefs) and emotions that can result in a change in behavior (behavioral change) that become habits through repetition.
4. line 156-158 Rephrase the sentence more appropriately. Does graphic content actually represent rational/emotional appeal or a set of it?
5. line 173. Question for the authors in terms of future work: Did you find literature data that artificial intelligence was used as a control instead of an independent researchers? How reliable was the cross-researchers control against the AI? If it has high correlation, it can be used as a good argument for your current study.
6. In the results section, I suggest showing appeals in relation to encourage and discouragement. This can be an interesting and important data.
7. In the results, I propose to show whether different media campaign typologies significantly differ in the frequency of using different appeals.
8. In the results, I propose to show which appeals are most often used together in relation to different media campaign typologies.
9. Pleasure is a very important emotion in the analysis of appeals. The concept of pleasure related to the dopamine system is important. Consider incorporating these emotional appeals. If the variable is not statistically suitable, discuss it adequately.
10. lines 414-415. This should be explained in more detail. Hypotheses that potentially explain this phenomenon should be stated.
11. lines 437-441. This should be explained in more detail. Hypotheses that potentially explain this phenomenon should be stated.
12. Think - How did the replacement of sugar with artificial sweeteners affect the effectiveness of campaigns? Did the same appeals and campains for the same product have the same effect or this replacement have some significant impact on product sale?
13. Explain the possible differences why different media campaigns use different appeals. This is is not enough explained in the text.
14. In future research, I recommend expanding the sample size. An international study that would include several types of campaigns from different regions would provide information on the existence of specific and non-specific perception of appeals.
The reviewer believes that the paper is well written and that the methodology is appropriate.
The conclusions are supported by the results obtained in the qualitative research. Recommendations for improvement are clearly laid out.
References are relevant and up to date.
Tables and figures are appropriately displayed.
Author Response

(The authors gave the same response as above.)

Round 2
Reviewer 1 Report
The authors successfully clarified the study aims in the abstract and introduction, making this paper's academic contribution much clearer. I appreciate that - as that was my primary concern with the paper originally.
That said, I'm disappointed that the authors didn't take more of my suggestions. Chiefly, I'm still struggling with the graphics findings. The authors coded the graphics sample across so many different dimensions but still really only interpret the emotional valence data (data also presented in and discussed as part of RQ2). Why might the coloring have varied across categories and/or what's the implication of the use of a differing color scheme (for example)? The lack of interpretation is especially frustrating because I spent a long time trying to follow the very lengthy Table 4--only to see that those findings were largely dropped in the back part of the paper. Is there more you can discuss? Or can you cut those findings if you aren't able or don't have the space to interpret them?
Assuming you keep them, is there more you can do to enhance ease of reading of Table 4? I disagree with your explanation that you're presenting qualitative data not suitable for chi-squares and similar (your coding scheme was dichotomous, and you used SPSS to analyze your results; so categories 1-4 strike me as suitable for quantitative analyses that go beyond descriptives). But if those kinds of analyses are beyond the scope of this paper, could you at least engage in parity in the narrative write-up in the Results alongside the table - summarizing the same codes for Category 1 and Category 4 (or all categories with sufficient data)? And as a smaller note about this Table/these analyses, it would be helpful if you included the n's for each category in the Table just for ease of reading/interpretation.
Secondary Comments:
- I still think that you should abridge the positive appeals section of the Discussion. I completely agree with you that that topic broadly is very important to this paper, but such an extensive focus on sub-dimensions you weren't able to code is unusual. The Discussion section of an empirical paper isn't the place for a lengthy review of research not reflected in the Results of that paper. I recommend either discussing positive appeals more generally, putting greater emphasis on the specific positive appeals that you were able to code, and/or shortening the discussion of the specific appeals that you couldn't code.
- Can you clarify what questions were left unanswered in study 2 in this sequence (reference #28)? It's unclear to me where that study leaves off and where the present paper begins - that is, what this paper, separate from the rest of the series, is contributing to the literature.
Minor Comments:
- "apeals" in the Note for Table 3 is a misspelling.
- "mied" on line 471 is also a misspelling.
Author Response
|
Reviewer 1 |
Authors’ Response |
Manuscript Revisions |
|
The authors successfully clarified the study aims in the abstract and introduction, making this paper's academic contribution much clearer. I appreciate that - as that was my primary concern with the paper originally. That said, I'm disappointed that the authors didn't take more of my suggestions. Chiefly, I'm still struggling with the graphics findings. The authors coded the graphics sample across so many different dimensions but still really only interpret the emotional valence data (data also presented in and discussed as part of RQ2). Why might the coloring have varied across categories and/or what's the implication of the use of a differing color scheme (for example)? The lack of interpretation is especially frustrating because I spent a long time trying to follow the very lengthy Table 4--only to see that those findings were largely dropped in the back part of the paper. Is there more you can discuss? Or can you cut those findings if you aren't able or don't have the space to interpret them?
|
We conducted chi-square analyses that assessed the relationships between the U.S. media campaign typology categories and graphic elements that revealed three variables with significant associations between the campaign typology and race/ethnicity, content, and product image. The chi-square test indicated significant difference at p < .05.
We added the results to the abstract though the current word count is 250. We defer to the editor to decide whether to cut this text in order to reach the word count limit of 200.
We also added a detailed description of the significant chi-square results on page 11 of the revised manuscript. In Table 4, we added the n= under each media typology category to enhance the interpretation, as requested.
In the discussion section on pages 14 and 15, we examined published literature about the use of colors and color combinations in advertising and marketing campaigns, even though this variable was not statistically significant in our study.
We hope that these revisions adequately address the reviewer’s comments.
|
Abstract (page 1) The frequent consumption of sugary beverages is associated with many health risks. This study examined how persuasive appeals and graphics were used in different media campaigns to encourage and discourage sugary beverages and water in the United States (U.S.) The investigators developed a codebook, protocol and systematic process to conduct a qualitative content analysis for 280 media campaigns organized into a typology with six categories. SPSS version 28.0 was used to analyze rational and emotional appeals (i.e., positive, negative, coactive) for campaign slogans and taglines; and graphic images (i.e., symbols, colors, audiences) for 60 unique campaigns across the typology. Results showed that positive emotional appeals were used more to promote sugary beverages in corporate advertising and marketing (64.7%) and social responsibility campaigns (68.8%), and less to encourage water in social marketing campaigns (30%) . In contrast, public awareness campaigns used negative emotional appeals (48.1%); and advocacy campaigns combined rational (30%) and emotional positive (50%) and negative appeals (30%). Public policy campaigns used rational (82.6%) and positive emotional appeals (73.9%) to motivate support or opposition for sugary beverage tax legislation. Chi-square analyses assessed the relationships between the U.S. media campaign typology categories and graphic elements that revealed three variables with significant associations between the campaign typology and race/ethnicity (χ2(103) = 32.445, p = .039), content (χ2(103) = 70.760, p < .001), and product image (χ2(103) = 11.930, p = .036). Future research should examine how positive persuasive appeals in text and graphics can promote water to reduce sugary beverage health risks.
Page 11 Table 4 shows the frequency of graphic and visual elements for the content analysis of a subset of 60 unique U.S. media campaigns used to encourage or discourage sugary beverages or encourage water and other healthy beverages to Americans across the six categories of the media campaign typology.
After calculating frequencies, chi-square analyses were used to assess the relationships between the U.S. media campaign typology categories and graphic elements. Analyses revealed three significant associations that included: campaign typology and race/ethnicity (χ2(103) = 32.445, p = .039), campaign typology and content (χ2(103) = 70.760, p < .001), and campaign typology and product image (χ2(103) = 11.930, p = .036).
Table 4 shows that corporate social responsibility, public relations or cause-marketing campaigns (81.8%), public information campaigns (67.7%), and social marketing campaigns (64.3%) were more likely to not depict people in the graphic images. Corporate marketing campaigns were more likely to depict white individuals (26.5%), whereas public information campaigns were more likely to depict individuals with unclear race/ethnic (16.2%) and black individuals (13.5%). Additionally, individuals were more likely to encounter positive messaging via corporate marketing campaigns (94.1%) and corporate social responsibility, public relations or cause-marketing campaigns (72.7%). Social marketing campaigns were more likely to present unclear content (72.7%), and public information campaigns were more likely to deliver negative content (48.6%).
Corporate marketing campaigns (79.4%), public information campaigns (70.3%), and media advocacy campaigns (60%) were more likely to feature a product image, whereas corporate social responsibility campaigns (63.6%) and social marketing campaigns (57.1%) were less likely to include a product image.
|
|
Secondary Comments: - I still think that you should abridge the positive appeals section of the Discussion. I completely agree with you that that topic broadly is very important to this paper, but such an extensive focus on sub-dimensions you weren't able to code is unusual. The Discussion section of an empirical paper isn't the place for a lengthy review of research not reflected in the Results of that paper. I recommend either discussing positive appeals more generally, putting greater emphasis on the specific positive appeals that you were able to code, and/or shortening the discussion of the specific appeals that you couldn't code. |
We did cut some extraneous text in the discussion section on pages 15-16 (please see the redline text removed).
However, we respectfully disagree with the reviewer’s suggestion to substantially cut the text related to the persuasive appeals that did not reach the ICR. After conferring, the senior authors decided to frame the discussion as future research needs. To provide some background to understand our decision, we searched the literature for months to identify articles and empirical studies to understand the distinct meanings to define various appeals to build our codebook that was refined over several months. A large portion of the literature on persuasive appeals was published during the 1990s and early 2000s (20-30 years ago). Marketing communications have evolved and become more sophisticated and integrated over this timeframe.
We did not find any recent comprehensive review of the literature for both rational and emotional appeals, especially that described media campaigns used to promote or discourage sugary beverage brands and products in the USA or any other country that are linked to obesity and many other health concerns. If the reviewer is aware of any recent review of persuasive appeals, please share it and we will cite it.
We believe that retaining the current text in the discussion will make an important contribution to the published literature to enable other researchers to examine this topic in greater depth to improve strategic communications that are increasingly on digital platforms and offer competing textual messages and graphic images. We defer to the editor to make the final decision. We believe that this paper has been strengthened by the external peer review process.
|
Page 15 4.1. Future Research on Positive Emotional Appeals: Pleasure, Happiness, Hope and Pride
Page 16 4.2. Future Research on Negative Emotional Appeals: Anger, Fear and Guilt
Page 17 4.3. How Findings May Inform Policies, Programs and Research to Promote Healthy Hydration |
|
Can you clarify what questions were left unanswered in study 2 in this sequence (reference #28)? It's unclear to me where that study leaves off and where the present paper begins - that is, what this paper, separate from the rest of the series, is contributing to the literature. |
We added text on page 4 in the background To clarify that the study 2 (Kraak et al. 2022) had identified six future research recommendations. Among these were the need to explore how message framing, graphics, slogans and IMC strategies may influence media campaign outcomes to promote healthy hydration and discourage sugary beverages.
There is a nice figure in the Kraak et al. 2022 publication that shows these six research recommendations that the reviewer can see in the open access publication at https://doi.org/10.1111/obr.13425
Kraak et al. 2022 did not examine persuasive appeals (i.e., rational, emotional or co-active) – only measured outcomes (i.e., cognitive, behavioral, policy and health) for a small proportion of the 280 media campaigns.
The analysis in this third paper submitted to the IJERPH addressed new research questions to build the evidence base in order to design more effective communications to socially normalize water and healthy hydration behaviors for Americans.
|
Page 4 Public health practitioners have used print, broadcast and digital media platforms to raise awareness and influence the behaviors of diverse populations to reduce sugary beverage health risks [27,28]. Different types of media campaigns operate concurrently within complex IMC ecosystems, yet the content of the appeals used in textual messages and graphic images are rarely analyzed [27,28]. Kraak et al. (2022) identified six future research recommendations, including the need to explore how message framing, graphics, slogans and IMC strategies may influence media campaign outcomes [28]. This study addresses this research gap to understand how persuasive appeals have been used in different types of media campaigns to design effective communication strategies that discourage sugary beverages and encourage water to reduce sugary beverage health risks for populations.
Ref 28. Kraak et al. How have media campaigns been used to promote and discourage healthy and unhealthy beverages in the United States? A systematic scoping review to inform future research to reduce sugary beverage health risks to Americans. Obes. Rev. 2022, 23, e13425. https://doi.org/10.1111/obr.13425
|
|
Minor Comments: - "apeals" in the Note for Table 3 is a misspelling. - "mied" on line 471 is also a misspelling. |
Thank you. We have corrected these words to “appeals” and “mixed” in the revised manuscript. |
|
